# Exploring Antimycobacterial Potential: Safety Evaluation and Active Compound Isolation from *Gymnopilus junonius*

**DOI:** 10.3390/antibiotics14020179

**Published:** 2025-02-11

**Authors:** Jenske Didloff, Gerhardt J. Boukes, Mutenta N. Nyambe, Denzil R. Beukes, Mookho S. Lerata, Velile Vilane, Michael Lee, Sharlene Govender, Maryna van de Venter

**Affiliations:** 1Department of Biochemistry and Microbiology, Nelson Mandela University, P.O. Box 77000, Gqeberha 6031, South Africa; s212278479@mandela.ac.za (J.D.); gerhardt.boukes@afrigen.co.za (G.J.B.); sharlene.govender@mandela.ac.za (S.G.); 2Afrigen Biologics (Pty) Ltd., c/o South African Medical Research Council, P.O. Box 19070, Tygerberg 7505, South Africa; 3Department of Pharmaceutical Chemistry and Pharmacognosy, Faculty of Pharmacy, Nutrition and Dietetics, Lusaka Apex Medical University, Lusaka 10101, Zambia; mutenta.nyambe@lamu.edu.zm; 4School of Pharmacy, University of Western Cape, Bellville 7535, South Africa; dbeukes@uwc.ac.za (D.R.B.); 3520938@myuwc.ac.za (M.S.L.); 5Center for High-Resolution Transmission Electron Microscopy (HRTEM), Physics Department, Nelson Mandela University, P.O. Box 77000, Gqeberha 6031, South Africa; velile.vilane@mandela.ac.za (V.V.); michael.lee@mandela.ac.za (M.L.)

**Keywords:** *Gymnopilus junonius*, antimycobacterial activity, gymnopilene, *Mycobacterium tuberculosis*, cytotoxicity

## Abstract

Background/Objectives: Tuberculosis remains a major public health crisis, and it is imperative to search for new antimycobacterial drugs. Natural products, including medicinal macrofungi, have been used as sources for the discovery of pharmaceuticals; however, research on their antimycobacterial activity remains limited. This study aimed to isolate and identify the bioactive compound responsible for antimycobacterial activity, thereby expanding on the limited knowledge regarding the antimicrobial activity and bioactive compounds present in *Gymnopilus junonius*. Methods: Bioassay-guided fractionation using column chromatography and preparative thin-layer chromatography were employed to isolate the active compound. Antimycobacterial activity against *Mycobacterium tuberculosis* H37 was assessed using a resazurin microplate assay (REMA). The chemical structure was determined by ^1^H nuclear magnetic resonance (NMR) spectroscopy, heteronuclear single quantum coherence (HSQC) spectroscopy, heteronuclear multiple bond correlation (HMBC) spectroscopy, and high-resolution electrospray ionization mass (HR-ESI-MS) spectrometry. Transmission electron microscopy (TEM) was used to observe the ultrastructural changes in *M*. *tuberculosis* induced by the compound. Cytotoxicity was evaluated in African green monkey kidney cells (Vero), human liver cells (C3A), and zebrafish embryos/larvae. Results: Bioassay-guided fractionation led to the isolation of gymnopilene, which showed inhibitory activity against *M*. *tuberculosis* (MIC: 31.25 µg/mL). TEM analysis revealed that treatment with gymnopilene caused ultrastructural damage observed as the disruption and disintegration of the cell wall. While gymnopilene demonstrated cytotoxicity in Vero and C3A cells, no toxicity was observed in zebrafish embryos/larvae for the crude extract. Conclusions: This study highlights that macrofungi, such as *G*. *junonius*, could be a valuable resource of bioactive compounds.

## 1. Introduction

Tuberculosis (TB) is an airborne and contagious disease that is caused by the bacillus *Mycobacterium tuberculosis*. It is a leading cause of death by a single infectious agent, and as per the World Health Organization (WHO) estimate, approximately 1.25 million individuals died from TB in 2023 [1]. Africa (24%), the South-East Asia Region (45%), and the Western Pacific Region (17%) were recognized as the WHO regions with the highest burden of disease [2].

TB is a curable and preventable disease; however, the inappropriate use and prescription of antimycobacterial drugs as well as poor treatment compliance have led to the spread of drug resistance [3,4]. Current treatment options have several drawbacks, which include long-term (~6 months) drug usage which can lead to drug-induced hepatotoxicity [4] and numerous other adverse effects including gastrointestinal intolerance, central nervous system toxicity, ocular toxicity, erythromelalgia, neuropathy, and skin rashes [5]. The treatment of drug-resistant strains is prolonged and requires the use of more expensive and toxic drugs that are less tolerable. Another challenge is co-infection with human immunodeficiency virus (HIV) which requires the careful selection of anti-TB and antiretroviral (ARV) drugs due to drug–drug interactions that can lead to drug-induced hepatitis [4].

Consequently, the search for new compounds with antimycobacterial properties is a priority to combat the TB problem [6,7], either for the development of new therapeutic agents or to be used to assist current treatment options [8]. Research has confirmed the promise of natural sources with antimycobacterial activity such as plants, microorganisms, and marine organisms [9]. However, natural products that exhibit antimycobacterial activity have not been extensively explored as potential lead compounds for TB drug discovery [10].

The genus *Gymnopilus* consists of gilled mushrooms within the Hymenogastraceae family, with worldwide distribution. Approximately 250 species have been described; however, only a few have been investigated for their chemical composition and biological activity. The species that have been most extensively investigated are those with hallucinogenic properties, owing to the production of psilocin and psilocybin [11,12].

The hallucinogenic mushroom *Gymnopilus junonius* (Fr.) P.D. Orton (previously classified as *G*. *spectabilis* (Fr.) Singer) is one of the most widely studied species (Figure 1). This mushroom is widely known as the big laughter mushroom because its consumption leads to uncontrolled laughter [13]. *G*. *junonius*, also known as the orange tuft, is an inedible bitter mushroom that causes brown rot in dead and dying trees [14,15].

Research on the antibacterial activity of *Gymnopilus* is limited, and only a few studies have screened extracts against selected Gram-negative and Gram-positive bacteria. A study by Nowacka et al. [8] demonstrated the antibacterial activity of an ethanol extract of *G*. *penetrans* against several bacterial species (e.g., *Staphylococcus epidermidis*, *Bacillus subtilis*, *Micrococcus luteus*, *Escherichia coli*, *Pseudomonas aeruginosa*) [8]. A methanol extract of *G*. *junonius* showed antibacterial activity against *Eggerthella lenta*, *Enterococcus faecalis*, and *Vibrio parahaemolyticus* [17]. Gas chromatography and mass spectrometry analyses of the extract identified 33 compounds, with octadecanoic acid, 2,3-dihydroxypropyl ester, 1,4:3,6-Dianhydro-alpha-d-glucopyranose, methyl stearate, benzeneacetic acid, phenol, 2,4-bis (1,1-dimethylethyl), 1H-Imidazole, 4,5-dihydro-4-methyl-2-phenyl-, octadecanoic acid, and 2-hydroxy-1-(hydroxymethyl) ethyl ester being the most prevalent compounds present [17].

Several studies have also reported on the cytotoxic properties of *G*. *junonius* [13,16]. Tremulane sesquiterpenes and gymnopilin K have been isolated and have shown cytotoxic effects on various cancer cell lines [13,18]. Additional bioactive compounds isolated from the *Gymnopilus* genus include gymnopilin A9, gymnopilin A10, gymnopilene, gymnoprenol A9 [18], and trichothecene and tremulane sesquiterpenes [13], which have demonstrated cytotoxic activity. Hispidin and bisnoryangonin have also been isolated and show antioxidant activity [19].

*G*. *junonius* was selected for bioactivity-guided fractionation based on promising antimycobacterial results obtained in a previous study [20,21]. The ethanolic extract screened against the *M*. *tuberculosis* H37 strain showed strong antimycobacterial activity as well as antibacterial activity against other respiratory bacterial pathogens [21]. Therefore, this study aimed to isolate and identify the bioactive compound responsible for antimycobacterial activity, thereby expanding on the limited knowledge regarding the antimicrobial activity and bioactive compounds present in *G*. *junonius*.

## 2. Results

### 2.1. Preliminary Antimycobacterial Screening of an Ethanol Extract of G. junonius

Preliminary antimycobacterial screening conducted on an ethanol extract of *G*. *junonius* revealed the inhibitory potential of the extract against *M*. *tuberculosis* H37, with a minimum inhibitory concentration (MIC) value of 62.5 µg/mL [20,21]. In the current study, the MIC of the extract prepared in 2018 from stored fruiting bodies was found to be 250 µg/mL using the resazurin microtiter assay (REMA). Due to the inability to obtain fresh fruiting bodies at the start of the study and based on the interpretation of antibacterial activity, where MIC values below 100 µg/mL are considered significant, and values between 100 and 625 µg/mL are considered moderate [22,23,24], the stored crude extract still exhibited moderate antimycobacterial activity. This encouraged further investigations to isolate and identify the bioactive compound responsible for the observed activity.

### 2.2. Bioassay-Guided Fractionation and Determination of the Antimycobacterial Activity of Isolated Fractions from a G. junonius Extract Using the Resazurin Microtiter Assay (REMA)

The ethanol extract was subjected to bioassay-guided fractionation using silica gel chromatography and gradient elution using n-hexane, ethyl acetate, and methanol as mobile phases. The fractions were subjected to thin-layer chromatography (TLC), and several compounds were visible under ultraviolet (UV) light at 254 nm. Several additional compounds were observed after viewing under UV light at 365 nm and staining with a vanillin–sulfuric acid solution. The compounds stained blue/purple and yellow/orange.

The antimycobacterial activity of isolated fractions 1–21 (Fr 1–21) was screened against *M*. *tuberculosis* H37 using the REMA method (Figure 2). The plates were visually examined to determine the MIC value, and inhibitory activity was quantified by fluorescence measurements (Ex/Em = 560/590). A change in color from blue to pink was interpreted as growth.

Fraction 13 showed the most promising results and inhibited the growth of *M*. *tuberculosis* by 95% or more at all concentrations tested (125–500 µg/mL). The percentage yield of fraction 13 was 9.4%, while for the crude extract it was 19.5%, as reported by Boukes et al. [16]. Fractions 14 and 15 also exhibited inhibitory activities with MIC values of 250 µg/mL and 500 µg/mL, respectively (Figure 2). The MIC value for the control drug, isoniazid, was 7.81 µg/mL.

The MIC of fraction 13 was found to be 62.5 µg/mL (Figure 3A), with an inhibitory percentage of 85.4%. The TLC profile of fraction 13 revealed the presence of multiple bands, appearing purple after vanillin–sulfuric acid staining (Figure 3C). Several predominant yellow bands were also noted after exposure to iodine vapor (Figure 3D).

### 2.3. Isolation of Sub-Fractions from Fraction 13 Using Preparative TLC and Determination of Antimycobacterial Activity

Fraction 13, which showed the highest inhibitory activity, was subjected to further purification using preparative TLC. The mobile phase solvent mixture was optimized to achieve optimal separation and resolution. Fourteen sub-fractions (Fr 13.1–13.14) were isolated from the preparative TLC plate. The isolated sub-fractions were spotted onto TLC plates to observe their purity and subjected to antimycobacterial screening to identify the active sub-fraction(s). The TLC profiling of the isolated sub-fractions showed that some fractions still contained multiple compounds.

The sub-fractions isolated from fraction 13 were screened for antimycobacterial activity, and through visual assessment and fluorometric analyses, several sub-fractions showed varying degrees of inhibitory activity. The percentage inhibition of *M*. *tuberculosis* growth ranged from 0 to 99.8% (Figure 4). Sub-fractions 13.6–13.8 showed the most promising activity with an MIC value of 31.25 µg/mL. The percentage inhibition of *M*. *tuberculosis* at the MIC value was 81.28%, 81.45%, and 64.80% for sub-fractions 13.6, 13.7, and 13.8, respectively. The sub-fractions 13.4 and 13.9 also showed inhibitory activity with MIC values of 62.5 µg/mL, while sub-fractions 13.5 and 13.10 showed an MIC value of 125 µg/mL.

### 2.4. Structural Elucidation

The TLC staining patterns of sub-fractions 13.6–13.8 displayed consistent profiles, along with similar antimycobacterial activities. Consequently, these sub-fractions were combined for structural elucidation and from now on will be referred to as sub-fraction 13.6.

A high-performance liquid chromatography–high-resolution electron spray ionization mass spectrometry (HPLC-HRESI-MS) analysis (Figure 5) of sub-fraction 13.6 revealed the presence of a prominent peak at *m/z* 825.7169 (M + H).

The ^1^H nuclear magnetic resonance (NMR) spectra of both the individual sub-fractions and combined sub-fractions 13.6–8 (F13.6) were acquired. The spectra obtained revealed their identical nature; however, they contained varying amounts of lipid impurity. These findings confirm the similarities observed between these fractions through TLC staining.

The ^1^H NMR of sub-fraction 13.6 (Figure 6) demonstrated a distinguishing feature with the presence of olefinic signals at δ_H_ 5.21, 5.06, and 5.91. Another olefinic signal was observed at δ_H_ 5.14 integrating two protons. The lipid impurity showed signals at δ_H_ 0.88, 2.34, 3.65, 3.93, and 4.18.

Heteronuclear single quantum coherence (HSQC) spectroscopy was used to determine the direct ^1^H–^13^C correlation of proton and carbons within the molecule. A HSQC spectrum is a 2D correlation map where the x-axis represents the ^1^H NMR spectrum and the y-axis represents the ^13^C spectrum. The expansion of the HSQC spectrum shows the resonances for the H1 (Figure 7A) and H2 (Figure 7B) vinyl protons, which are protons directly adjacent to the carbon–carbon double bond. Figure 8 shows the resonances for the aliphatic region.

Heteronuclear multiple bond correlation (HMBC) spectroscopy was performed to identify proton–carbon signals that are coupled over two to four bonds apart. The expansion of the HMBC NMR spectrum in Figure 9 shows resonances for the vinyl protons, while Figure 10 shows resonances for the aliphatic region.

Table 1 indicates the chemical shift values (δ values) for all the carbons and protons of the predominant compound present in sub-fraction 13.6. Carbon nuclei were assigned using HSQC and HMBC spectroscopy. The predominant compound present in sub-fraction 13.6 was identified as gymnopilene.

### 2.5. TEM

Transmission electron microscopy was performed to evaluate the changes induced in the cell morphology of *M*. *tuberculosis* after treatment with sub-fraction 13.6 containing the predominant compound gymnopilene isolated from *G*. *junonius*.

The untreated *M*. *tuberculosis* cells displayed well-conserved cell morphology with characteristic features including the cell envelope, the nucleoid region, and the presence of ribosomes. The cell envelope, shown in Figure 11D, is composed of an electron-dense outer membrane (OM), an electron transparent layer (periplasm), and an inner membrane (IM). The length of the cells ranged between 1.2 and 1.4 µm with an average cell diameter of 341 ± 13.6 nm (Figure 11A–C).

*M*. *tuberculosis* treated with isoniazid, a known cell wall inhibitor, showed ultrastructural changes compared to the untreated cells. There were clear signs of cell wall disintegration with a lack of inner and outer membrane layers in some areas, as highlighted by the black arrows (Figure 12). The cytoplasmic content also appeared to be altered, being more electron-dense in the center compared to the untreated cells (Figure 12).

The treatment of *M*. *tuberculosis* with sub-fraction 13.6 containing the predominant compound, gymnopilene, from *G*. *junonius* also showed cell wall damage (Figure 13A–F) as well as cell wall disintegration (Figure 13G–I).

### 2.6. Cytotoxicity Against Human Liver Cells (C3A) and African Green Monkey Kidney Cells (Vero)

The cytotoxic effects of the crude extracts, isolated fractions, the predominant compound gymnopilene, and the known cytotoxic drug melphalan were screened against the C3A and Vero cell lines. Cell numbers (live and dead) were determined by Hoechst 33342 and propidium iodide (PI) dual staining following 48-h treatments.

A drastic decrease in the percentage of live cells (C3A) after treatment with crude macrofungal extract, fraction 13, and sub-fraction 13.6 containing gymnopilene was observed, suggesting an increase in toxicity with increasing concentration (Figure 14A). The *G*. *junonius* extract caused a significant dose-dependent decrease in the number of live cells at all concentrations tested (*p* ≤ 0.005). Fraction 13 (at all concentrations) and gymnopilene (12.5–50 µg/mL; *p* ≤ 0.005) isolated from *G*. *junonius* also decreased the number of live cells. The positive control, melphalan, demonstrated a dose-dependent decrease in cell viability. Isoniazid showed a significant decrease from 2.5 mM.

The treatment of Vero cells (Figure 14B) with the crude ethanol extract of *G*. *junonius* resulted in a dose-dependent decrease in the number of live cells. A significant decrease was observed at 25 µg/mL (*p* ≤ 0.01) and 50–100 µg/mL (*p* ≤ 0.005). Fraction 13 and sub-fraction 13.6 containing gymnopilene showed toxic effects at concentrations ≥12.5 µg/mL (*p* ≤ 0.005) and ≥25 µg/mL (*p* ≤ 0.005). A significant increase in the number of dead cells was observed for *G*. *junonius* fraction 13 and gymnopilene at 50 µg/mL.

The IC_50_ values were determined for each cell line from dose–response curves of live cell numbers (Table 2). The crude *G*. *junonius* extract yielded an IC_50_ value of 90.86 ± 1.04 µg/mL against the Vero cell line. When isolated fraction 13 and gymnopilene were compared to the crude extract, both these treatments showed higher cytotoxicity, with IC_50_ values of 27.02 ± 1.01 µg/mL and 28.53 ± 1.02 µg/mL, respectively. The extract, isolated fraction, and gymnopilene showed more toxicity to C3A cells compared to Vero cells, with IC_50_ values of 18.93 ± 1.09 µg/mL, 9.16 ± 1.05 µg/mL, and 22.38 ± 1.09 µg/mL, respectively.

The selectivity index was calculated using the IC_50_ values against the selected cell lines and the MIC value obtained against *M*. *tuberculosis*. The selectivity indices for all the treatments were found to be <1.

### 2.7. Hepatotoxicity Assessment

Reactive oxygen species (ROS) production was investigated as a marker to identify the possible risk of drug-induced hepatotoxicity. This was determined using the CellRox^®^ orange staining method after treatment with the extract, fraction, and sub-fraction 13.6 (gymnopilene) (Figure 15). Excessive cytotoxicity lowers the cell density to a level where accurately assessing morphological features, vital for predicting hepatotoxicity and genotoxicity, becomes challenging [26]. Therefore, this study focused on the two to three lowest concentrations of each treatment (below or equal to the IC_50_ value), where the cell viability was ≥50%, to ensure that the interpretation of the results is reliable.

Treatment with the crude extract, isolated fraction, and sub-fraction 13.6 (gymnopilene) from *G*. *junonius* did not significantly increase ROS levels. The only significant increase observed for F13 isolated from *G*. *junonius* was noted at higher concentrations where cell viability was between 10 and 25%. As mentioned above, due to the significant reduction in cell density, the production of ROS could not be accurately assessed.

Melphalan showed significant increases in ROS production at 50 µM. The antimycobacterial drug, isoniazid, did not show an increase in ROS production at any of the concentrations tested.

Another mechanism through which drugs induce hepatotoxicity is mitochondrial dysfunction. To evaluate the effect on mitochondrial function, this study assessed mitochondrial membrane potential using Tetramethylrhodamine ethyl ester (TMRE) staining and quantified the mitochondrial content using MitoTracker Green (MTG) staining (Figure 16A,B). As shown in Figure 16A, exposure to the crude extract of *G*. *junonius* at concentrations ≤25 µg/mL did not yield a significant alteration in membrane potential compared to the control. Fraction 13 (IC_50_: 9.16 µg/mL) showed a small yet significant increase in membrane potential at 6.25 µg/mL (~5%), while no significant difference was observed at 12.5 µg/mL. Gymnopilene (IC_50_: 22.28 µg/mL) showed no notable change in membrane potential within the concentration range of 3.13–12.5 µg/mL; however, significant membrane depolarization was observed at 25 µg/mL (*p* ≤ 0.005). The mitochondrial content after treatment with the crude extract, isolated fraction, and gymnopilene showed dose-dependent increases (Figure 16B).

Treatment with the positive control, carbonyl cyanide m-chlorophenylhydrazone/CCCP (25 µM), displayed a significant decrease in the mitochondrial membrane potential (MMP) (*p* ≤ 0.005) and mitochondrial content (*p* ≤ 0.005). Melphalan (50 µM) was shown to cause mitochondrial membrane depolarization and increased the mitochondrial content. The antimycobacterial drug, isoniazid, at 10 mM showed membrane depolarization and increased the mitochondrial content at concentrations ≥2.5 mM.

### 2.8. Zebrafish Toxicity Screening

The in vivo effects of the macrofungal extracts were screened using the zebrafish embryotoxicity test. The zebrafish were evaluated at 24-, 48-, and 72-h post exposure (hpe) for general morphological development and the development of malformations. A total of 15 zebrafish embryos were used in this study per treatment condition. The viability of the zebrafish embryos/larvae was confirmed by the presence or absence of a heartbeat and blood circulation.

The development of the zebrafish embryos/larvae was assessed as described by Hermsen et al. [27] and Beekhuijzen et al. [28]. If all developmental hallmarks were present, each control/treatment group containing five zebrafish embryos/larvae could yield a maximum score of 60 points for general morphology (GMS). Analyses of the developmental toxicity of the ethanol extract of *G*. *junonius* from 24 to 72 hpe demonstrated no significant differences in the number of live zebrafish embryos/larvae compared to the control (Figure 17A). The treatment also did not alter the hatching rate of the embryos (Figure 17B). The cumulative general morphological score showed slight decreases after 48 and 72 hpe at 60–80 µg/mL.

Developmental impairments or malformations were also analyzed including malformations of the head, sacculi/otoliths, tail, and heart, yolk deformation, and a deformed body shape (Figure 18). An evaluation of teratogenic effects after exposure to the crude *G*. *junonius* extract (Figure 18) showed that only one or two zebrafish larvae displayed malformation of the tail, a deformed body shape, malformation of the heart (pericardial edema), and yolk deformation (yolk edema). The most prevalent developmental abnormality was the malformation of the tail at a concentration of 50 µg/mL, which increased from one larva to four larvae after 72 hpe. Most of the observed malformations did not appear to be concentration-dependent.

## 3. Discussion

Tuberculosis remains one of the infectious diseases that continues to be of global importance with high morbidity and mortality rates. The current therapeutic options available for the treatment of tuberculosis are also not adequate, with various adverse effects that limit patient compliance. The spread of drug-resistant strains has also hampered effective treatment. Therefore, there is a need to search for alternative therapeutic options. Natural resources have played an important role in drug discovery throughout the years. Macrofungi/mushrooms remain a largely unexplored source, and therefore, a vast amount of undiscovered knowledge exists.

To address these concerns, the aim of this study was to isolate, identify, and characterize the antimycobacterial compound present in the ethanol extract of *G*. *junonius* as well as to investigate its mechanism of action and potential toxicity using in vitro and in vivo screening methods.

Bioactivity-guided fractionation using column chromatography and TLC was performed to isolate the antimycobacterial compound present in an ethanol extract of *G*. *junonius* collected in Western Cape, South Africa. Three fractions (Fr 13–15) demonstrated the most promising results in terms of antimycobacterial activity. Fraction 13 exhibited > 95% inhibitory activity at concentrations ranging between 125 and 500 µg/mL. The MIC value of fraction 13 was determined to be 62.5 µg/mL. Fractions 14 and 15 showed MIC values of 250 and 500 µg/mL, respectively, suggesting lower levels of antimycobacterial activity. Based on the MIC values obtained, fraction 13 was identified as the most promising candidate for further purification. The preparative TLC of fraction 13 resulted in the isolation of 14 sub-fractions (Fr 13.1–13.14). Among these, sub-fractions 13.6–13.8 showed the most promising results, demonstrating antimycobacterial activity with an MIC value of 31.25 µg/mL.

Studies have identified several compounds produced by mushrooms that exhibit antimicrobial properties including terpenes, sesquiterpenes, benzoic acid derivatives, steroids, quinolines, proteins, and peptides [29,30]. The TLC profiles revealed that the extract contained a complex mixture of compounds isolated in the various fractions, which exhibited different retention factors (Rfs) and colors after staining with a vanillin–sulfuric acid solution. The predominant compounds present in the initial fractions based on the observed colors include terpenoids (blue or purple) [31,32] and flavonoids (yellow or orange) [33,34]. Another study stated that terpenoids could also stain blue, dark green, brownish-green, bluish-green, light orange, purple, lavender, maroon, and brown [35].

The active antimycobacterial components in sub-fractions 13.6–13.8 showed a distinct violet/purple color after staining with the vanillin–sulfuric acid solution (Figure 3). Iodine vapor is often used for the visualization of organic compounds as it has a high affinity for aromatic and unsaturated compounds. The exposure of terpenoid compounds to iodine vapor results in the appearance of yellow zones due to the conjugated double bonds in the structure (Figure 3D). The active components showed both purple and yellow bands after vanillin–sulfuric acid staining and exposure to iodine vapor, respectively, suggesting that they are terpenoid-type compounds.

Terpenoids are secondary metabolites which have been shown to be effective against several diseases, including microbial, viral, cancer, and neurodegenerative diseases [36]. Terpenoids including diterpenes and triterpenes have been isolated from various mushroom genera which have exhibited antimicrobial properties [37]. It is also well documented that several terpenoids isolated from mushrooms exhibit antimycobacterial activity. Lanostane triterpenes isolated from an ethanol extract of *Astraeus pteridis* exhibited antimycobacterial activity against *M*. *tuberculosis* H37Rv with MIC values ranging from 34 to 64 µg/mL [38]. The lanostane triterpenes astraodoric acids A and B, isolated from *A*. *odoratus*, showed moderate activity against *M*. *tuberculosis* H37Ra, with MIC values of 50 and 25 µg/mL, respectively [39]. *Ganoderma orbiforme* has also been a source of lanostane triterpenes (Ganorbiformins A–G) and other known compounds such as ganoderic acid T and its C-3 epimer. The majority of the isolated compounds were inactive or weakly active, whereas ganoderic acid T (MIC: 10 µM) and its C-3 epimer (1.3 µM) showed good activity against *M*. *tuberculosis* H37Ra [40].

The active sub-fractions of *G*. *junonius* (Fr 13.6–13.8) were subjected to 1D (^1^H NMR) and 2D NMR spectroscopy (HSQC, HMBC), and HPLC-HRESI-MS. The ^1^H NMR spectrum (Figure 6) of sub-fraction 13.6 revealed three olefinic signals at δH 5.21, 5.06, and 5.91, suggesting unsaturated functionalities. The latter mentioned signals are characteristic of a monosubstituted vinyl group. The resonances of the vinyl protons, H1 and H2, and the aliphatic region were identified upon the expansion of the HSCQ and HMBC spectra. An isoprenol moiety (-CH_2_-C(OH)-CH=CH_2_) was deduced from the combined HSQC and HMBC NMR spectral data. Furthermore, an additional olefinic signal at δH 5.14, integrating two protons, was observed, suggesting the presence of two trisubstituted double bonds. The information obtained from the ^1^H NMR and 2D NMR spectroscopic data indicated that the main compound in sub-fraction 13.6 is a polyisoprenepolyol. The data obtained were compared with the available NMR data in the literature, and the structure was identified as gymnopilene, which validated the polyisoprenepolyol structure. Finally, HPLC-HRESI-MS analysis revealed a prominent peak at m/z 825.7169 (M + H) consistent with the structure of gymnopilene (C_50_H_96_O_8_).

Gymnoprenols are compounds characterized by a distinctive polyisoprenepolyol structure containing 45–60 carbon atoms [41]. Gymnopilene was initially isolated by Findlay and He [25], and their study also noted the presence of a multiplet at δH 5.10 ppm and a vinyl moiety attached to C-3, similar to that observed in the current study. The reported bioactivities for gymnopilene are limited to its cytotoxic effects against several cancer cell lines, as demonstrated by Kim et al. [18]. To the best of the authors’ knowledge, there are no reports documenting the antimycobacterial activity of gymnopilene.

Upon further analysis of the lipid impurity within the fraction using NMR spectra, it was suggested to be a saturated monacylglycerol lipid. In a study by Suh et al. [42], a monoacylglycerol, 1-O-(6,6-dimethoxyhexanoyl)-glycerol, was isolated from a methanol extract of *G*. *spectabilis*. This monacylglycerol exhibited antiproliferative effects against a skin melanoma cell line (SK-MEL-2) and demonstrated inhibitory effects on nitric oxide production in LPS-stimulated BV-2 cells [42].

The screening of extracts and the isolation of compounds are important during drug discovery; however, it is also essential to investigate their potential mechanisms of action. The effect of gymnopilene on the internal structures of *M*. *tuberculosis* was investigated using TEM. Untreated *M*. *tuberculosis* cells showed well-conserved cell morphology with characteristic features including the cell envelope, the nucleoid region, and the presence of ribosomes (Figure 11). The cell structures observed were consistent with the findings of Yamada et al. [43], with the presence of an outer membrane (mycomembrane), periplasm, and inner membrane/plasma membrane.

*M*. *tuberculosis* treated with isoniazid showed clear signs of cell wall disintegration with a lack of inner and outer membrane layers in some areas (Figure 12). The cytoplasmic content also appeared to be altered, being more electron-dense in the center compared to the untreated cells. Similar cell wall damage was seen by Jyoti et al. [44] and Carcamo-Noriega et al. [45]. It can be observed from the TEM micrographs that treatment with sub-fraction 13.6 containing gymnopilene caused cell wall damage as well as cell wall disintegration (Figure 13). These observations suggest that the antimycobacterial activity of gymnopilene could be due to the disruption of membrane integrity, leading to cell death.

Several studies have isolated triterpenes and triterpenoids from various mushroom species which exhibit antimycobacterial activity [38,39,46]. However, the mechanism by which these compounds exert their antibacterial activity has not yet been investigated. Terpenoids are among the main components of natural products with bioactive properties [47]. These phytochemicals have shown five mechanisms by which they exert their antimicrobial activity, including cell membrane destruction, anti-quorum sensing action, the inhibition of ATP, the inhibition of protein synthesis, and synergistic effects. Cell membrane destruction occurs due to the lipophilic characteristics of terpenoids, which enable them to interact with and pass through the phospholipid bilayer. The damage caused by terpenoids affects the integrity of the cell membrane and the normal physiological activities which result in their antimicrobial activity [48].

The cell envelope of *M*. *tuberculosis* is a complex structure composed of several crucial layers, including mycolic acid, arabinogalactan, and peptidoglycan layers. The TEM micrographs revealed that the membrane was affected by the treatment; however, the exact target on the membrane structure could not be determined using transmission electron microscopy alone. Further investigation is required to determine the precise target and how the treatment affects the enzymes involved in the synthesis of each layer.

Treatment regimens for tuberculosis are associated with adverse side effects. Therefore, toxicity evaluation is an important component during drug discovery to identify potential drug candidates with minimal adverse effects. In this study, toxicity was assessed against Vero and C3A cells. Treatment with the extract, isolated fraction, and sub-fraction containing gymnopilene significantly decreased viable cells, indicating cytotoxic effects. The selectivity index (SI) value obtained was shown to increase during the fractionation process with the crude extract of *G*. *junonius* to yield fraction 13 and gymnopilene. Gymnopilene showed a significant increase in the SI, ~2.5 and ~9.5 times higher against Vero and C3A cells, respectively, compared to the crude extract. However, it is important to note that the SI values obtained ranged between 0.076 and 0.72, which indicates relatively low safety margins. This suggests that the extracts, fractions, and gymnopilene exhibited higher toxicity toward the cell lines compared to *M*. *tuberculosis*.

Numerous mushrooms are recognized for their cytotoxic effects and are renowned for their anticancer properties. Among these mushrooms are *Agaricus blazei*, *Cordyceps militaris*, *Fomitopsis lilacinogilva* (currently known as *R*. *lilacinogilva*), *Ganoderma lucidum*, *Gymnopilus junonius* (synonym: *G*. *spectabilis*), *Hericium erinaceus*, *Inonotus obliqus*, *Pycnoporus sanguineus*, and several others. The observed anticancer effects are due to the presence of cytotoxic compounds, including ganoderic acid, ergosterol, ergosterol peroxide, hispidin, lanostanes, psilocybin, linolenic acid, and various other bioactive compounds [49].

A study by Kim et al. [18] isolated several compounds from a methanol extract of *G*. *spectabilis* (synonym *G*. *junonius*) including gymnopilene, gymnopilin K, gymnopilin A9, gymnopilin A10, and gymnoprenol A9. The study found that these compounds exhibited cytotoxicity toward several cancer cell lines (A549, SK-OV-3, SK-MEL-2, and HCT-15). Gymnopilene showed cytotoxic effects against the selected cell lines with IC_50_ values ranging between 14.40 and 24.10 µM [18]. Additionally, a monoacylglycerol was isolated from *G*. *junonius*, exhibiting cytotoxicity against SK-MEL-2 with an IC_50_ value of 26.43 µM [42]. Furthermore, sesquiterpenes from *G*. *junonius* demonstrated cytotoxic effects on human lung and prostate cancer cell lines [13].

Hepatotoxicity can be caused by several mechanisms such as mitochondrial dysfunction, oxidative stress, DNA damage, protein modification, lipid peroxidation, and the failure of repair mechanisms. The mechanisms of hepatotoxicity may overlap with each other, which can lead to a cascade of events [50]. To further assess the potential risk of drug-induced hepatotoxicity posed by the extracts, isolated fractions, and gymnopilene, this study examined changes in ROS levels and mitochondrial function in C3A cells. Through the examination of these parameters, a better prediction of the effects that the treatments have on liver cells could be made.

The potential risk of drug-induced hepatotoxicity was also assessed by investigating ROS production and mitochondrial function in C3A cells. These parameters were only assessed at concentrations below the IC_50_ concentration obtained. It was observed that treatment with the extract, fraction, and sub-fraction containing gymnopilene did not increase ROS production. Mitochondrial function was evaluated based on the effect of treatment on MMP and mitochondrial content. The crude extract and sub-fraction containing gymnopilene did not have a significant effect on MMP. Fraction 13 showed a significant increase in MMP at 6.25 µg/mL, although this effect was not observed at 12.5 µg/mL. The mitochondrial content was shown to increase at the relevant concentrations assessed for the extract, fraction, and gymnopilene.

Based on the results obtained, it appears that mitochondrial toxicity due to ROS production seems to be an unlikely mechanism. The observed increase in mitochondrial content following treatment with the extract, fraction, and gymnopilene could be due to the induction of cellular stress and damage to the mitochondria. In response to this, cells often increase mitochondrial biogenesis as a compensatory mechanism in which the cells try to maintain mitochondrial function by increasing the number of mitochondria. This is often carried out to maintain redox homeostasis and energy demands [51,52]. These findings indicate a potential mitotoxic effect of the extract, fraction, and gymnopilene. However, mitochondrial dysfunction can involve various targets, which can lead to a cumulative effect. Therefore, to fully elucidate the mechanism of hepatotoxicity, alternative targets should be investigated.

To further assess the safety of the crude extract, an in vivo zebrafish embryo model was employed. Exposure to the *G*. *junonius* extract showed no lethal effects as no significant decrease in the number of live zebrafish embryos/larvae was observed over the 72 hpe treatment period. The hatching ability of the zebrafish was also not affected by the treatment with *G*. *junonius*.

Several zebrafish embryos/larvae, including those in the control treatment group, showed developmental abnormalities. The total number of zebrafish displaying these malformations was typically one or two embryos/larvae. The results obtained therefore show that there is no distinct correlation between the treatment concentration and the number of malformations observed (Figure 18). The only time-dependent observation made was at a concentration of 50 µg/mL, where an increase in the number of zebrafish with tail malformations was noted (Figure 18A).

The in vivo toxicity results suggest that the *G*. *junonius* extract appears to be non-toxigenic. These findings for *G*. *junonius* contradict the in vitro cytotoxicity screening where significant toxicity was observed against both the Vero and C3A cell lines. This highlights the importance of using different models to assess toxicity. Several factors could contribute to the observed differences in toxicity for *G*. *junonius* such as. the inability of the compound to be taken up by zebrafish embryos/larvae, or because the pathway associated with the observed toxicity in cell lines is absent in zebrafish. Additionally, the concentration to which zebrafish embryos/larvae are exposed might also be lower than that of the cells, resulting in reduced toxicity. These factors should be investigated to ensure that the deduction made is accurate.

In conclusion, this study demonstrated that gymnopilene, a compound isolated from an ethanol extract of *G*. *junonius*, exhibited antimycobacterial activity against *M*. *tuberculosis* with an MIC of 31.25 µg/mL. Gymnopilene was shown to disrupt and disintegrate the bacterial cell wall, a mechanism similar to isoniazid, one of the current TB drugs given to patients. While the crude extract and isolated fractions showed cytotoxic effects in mammalian cells, the crude extract did not show toxicity in zebrafish embryos/larvae. Several limitations were identified in the study, which could be addressed in future research. These include the inability to obtain fresh fruiting bodies of *G*. *junonius* at the start of this study. The usage of a preparative TLC method to extract the active sub-fractions might not be optimal, and alternative purification methods could be explored to improve quantity and purity. Additionally, this study focused solely on the bacterial cell wall as a potential target, so further investigation of alternative mechanisms should be conducted. Future investigation should focus on elucidating the therapeutic potential, safety profile, and broader mechanism of action of gymnopilene. Additionally, the exploration of other macrofungi for bioactive compounds holds considerable promise in identifying novel agents that could be developed for the treatment of tuberculosis.

## 4. Materials and Methods

### 4.1. Macrofungal Collection, Identification, and Extract Preparation

Fruiting bodies of *Gymnopilus junonius* (Fr.) P.D. Orton were collected in Plettenberg Bay/Knysna (Harkerville, −34.047376, 23.246311), Western Cape, South Africa. Identification was performed based on morphological characteristics using South African mushroom guides [14,15] and confirmed by Mushroom Guru (Pty) Ltd. (Somerset West, South Africa). A specimen was deposited in the Nelson Mandela University (Gqeberha, South Africa) herbarium (PEU 25354).

The ethanol extract was prepared as previously described by Boukes et al. [16]. Briefly, macrofungal fruiting bodies were dried in an oven at 25–30 °C for 2–3 days. Once dried, the fruiting bodies were submerged in liquid nitrogen and crushed using a mortar and pestle. An 80% ethanol solution was added to the powdered material at a ratio of 1:15 (*w*/*v*) and stirred continuously at room temperature for 24 h. The mixture was centrifuged at room temperature at 1800× *g* for 5 min using the Eppendorf Centrifuge 5810 (Hamburg, Germany). The supernatant was filtered twice through Whatman No. 1 filter paper under vacuum. The ethanol was subsequently evaporated using the BUCHI Rotavapor R-210 (Flawil, Switzerland) at 50 °C. The resulting extract was frozen at −80 °C prior to freeze-drying with a VirTis SP Scientific Sentry 2.0 freeze-dryer (Gardiner, NY, USA). The macrofungal extract was stored in a desiccator at 4 °C in the dark until further use.

### 4.2. Bioactivity-Guided Fractionation of Ethanol Extract

The fractionation and isolation of the antimycobacterial compound(s) from the *G*. *junonius* ethanol extract was carried out as illustrated in the fractionation scheme in Figure 19.

#### 4.2.1. Silica Gel Chromatography

The ethanol extract was subjected to silica gel (0.063–0.200 mm, Merck, Germany) column chromatography using a gradient solvent system of increasing polarity with hexane/ethyl acetate (EtOAc) mixtures, followed by EtOAc/methanol (MeOH) mixtures (Table 3). The chromatography and TLC protocols were adapted from Venables et al. [53] and Gini and Jeya Jothi [54].

A glass column (30 cm × 1 cm) was prepared for silica gel column chromatography by rinsing the column with hexane and allowing it to dry. The silica gel was then packed into the column as a slurry using hexane, achieving a column length of approximately 14.5–15 cm. The macrofungal extract was adsorbed onto silica at a 1:2 extract-to-silica ratio by dissolving it in 100% methanol. After the methanol was evaporated, the extract was loaded onto the prepared column, resulting in a band with a width of ±15 mm.

The macrofungal extract was fractionated using gradient elution with hexane/EtOAc and EtOAc/MeOH as mobile phases (Table 3). Fractions of 10 mL were collected in pre-weighed tubes, and the solvent was evaporated using a Büchi rotary evaporator (50 °C) or under vacuum. After solvent evaporation, the tubes were weighed, and the masses of each fraction were recorded.

Each fraction was analyzed by TLC on pre-coated 60 F254 silica gel aluminum plates (Merck, Merck, Germany). The fractions were re-dissolved in their respective elution solvents at a concentration of 10 mg/mL and then loaded onto a 10 × 10 cm TLC plate and allowed to air-dry. The chromatogram tank was equilibrated for 20 min with an appropriate liquid solvent system and developed until the solvent front reached approximately 1 cm from the top of the plate. After air-drying, the TLC plates were visualized under UV light (λmax = 254 and 365 nm) and then sprayed with a vanillin–sulfuric acid solution (0.1 g vanillin, 28 mL ethanol, and 1 mL sulfuric acid) and heated to induce color formation. The migration of the compounds was assessed, and their retention factor (Rf) values were recorded. Photographs of the developed TLC plates were captured. Additionally, the fractions were evaluated for antimycobacterial activity using the resazurin microtiter assay (REMA). Fractions exhibiting the highest inhibitory activity against *M*. *tuberculosis* and the lowest MIC values were selected for further purification via preparative TLC.

#### 4.2.2. The Determination of the Antimycobacterial Activity of the Crude Extract and Fractions Using REMA

The crude extract and fractions obtained from column chromatography were screened using a whole-cell-based approach to identify those with activity against the *Mycobacterium tuberculosis* H37 strain. A culture of *M*. *tuberculosis* was prepared by inoculating a 10 mL Middlebrook 7H9 broth base (Thermo Fischer Scientific, Waltham, MA, USA) supplemented with 10% oleic acid, albumin, dextrose, catalase supplement base (Thermo Fischer Scientific, Waltham, MA, USA), 0.2% (*v*/*v*) glycerol, and 0.05% (*v*/*v*) Tween 80 (Sigma-Aldrich, St. Louis and Burlington, MA, USA). The culture was grown for 10 days at 37 °C.

The susceptibility of *M*. *tuberculosis* to the extract and fractions was assessed using the REMA in a 96-well microtiter plate as previously described by Franzblau et al. [55] with minor modifications. To prevent evaporation during incubation, sterile water was added to the perimeter wells. Plate layouts and concentration ranges may have differed with experiments, but the dilution protocol remained consistent.

Briefly, 50 µL of the culture broth was added to all experimental wells except the third column, which contained the initial concentration of extracts, fractions, or compounds. The second column served as a sample control. In the third column, 100 µL of the sample was added and twofold serial dilution was performed by transferring 50 µL from each well to the next, mixing thoroughly between each transfer. Serial dilutions were prepared as follows—crude extracts: 125–500 µg/mL; and *G*. *junonius* fractions: 15.13–125 µg/mL. This process continued until the well containing the last concentration was reached, and 50 µL was discarded. The *M*. *tuberculosis* inoculum was prepared by adjusting the culture grown for 10 days to a 0.5 McFarland standard (absorbance at 600 nm = 0.08–0.1; equivalent to ~1.5 × 10^8^ cells/mL), using a Middlebrook 7H9 broth medium. The suspension was well vortexed, and 50 μL of the standardized inoculum was added. 

The controls included a growth control (50 μL culture broth + 50 μL *M*. *tuberculosis* inoculum), a dimethyl sulfoxide (DMSO) control (50 μL culture broth and 50 μL 8% DMSO), a broth control (100 μL culture broth), and a sample control (50 μL medium + 50 μL of the highest concentration of extract, fraction, compound, or control antibiotic, i.e., isoniazid). The plates were sealed and incubated at 37 °C for 7 days.

After incubation, 12.5 µL of 20% Tween 80 and 20 µL of resazurin blue dye (Sigma-Aldrich, St.Louis and Burlington, MA, USA) were added to each well. Following additional 6-h incubation at 37 °C, a color change from blue to pink indicated bacterial growth. The biohazard risk of *M*. *tuberculosis* aerosol transmission was eliminated by adding formaldehyde to reach a final concentration of 10%. Fluorescence was measured using a BioTek^®^ SYNERGY Mx fluorometer (Winooski, Winooski, VT, USA) at an excitation wavelength of 560 nm and emission wavelength of 590 nm. The MIC was defined as the lowest concentration that prevented the color from changing from blue to pink. The percentage inhibition of the fractions was calculated using the equation below [56]. All experiments were performed in triplicate and repeated three independent times unless otherwise stated.$$\mathrm{Percentage}\;\mathrm{inhibition}=\left[1-\left(\mathrm{Test}\;\mathrm{well}\;\mathrm{Relative}\;\mathrm{Fluorescence}\;\mathrm{Units}\;\left(\mathrm{RFUs}\right)/\mathrm{Mean}\;\mathrm{RFUs}\;\mathrm{growth}\;\mathrm{control}\;\mathrm{wells}\right)\right]\times 100$$

#### 4.2.3. Isolation of Active Antimycobacterial Compound from *G. junonius* Fraction Using Preparative TLC

Preparative TLC was performed to further purify the fraction exhibiting antimycobacterial activity. The active fraction was dissolved in the same solvent used during column chromatography and applied to a 20 × 20 cm pre-coated 60 F254 silica gel aluminum plate (Merck, Germany). A large chromatogram tank was equilibrated for 30 min using a 9:1 (*v*/*v*) ethyl acetate/methanol solvent system with 5 drops of ammonium hydroxide, which was used for the further chromatographic separation of fraction 13.

The plate was developed until the solvent front reached approximately 1 cm from the top of the plate. After development, the plate was air-dried, and the separated bands were identified by UV light at 254 nm and 365 nm. One section of the preparative TLC was stained with a vanillin–sulfuric acid solution. The individual bands were identified, and the bands were scraped from the TLC plates and eluted from the silica using warm methanol, yielding sub-fractions. These sub-fractions were filtered using cotton wool and rinsed once with 1 mL methanol and collected in dry, pre-weighed tubes. The solvent was evaporated using a Büchi rotary evaporator (50 °C) or under vacuum. Isolated sub-fractions were subsequently monitored on 10 × 10 cm TLC plates and screened for antimycobacterial activity using the REMA as described in Section 4.2.2.

#### 4.2.4. HPLC-HRESI-MS Acquisition

High-performance liquid chromatography was combined with quadrupole time-of-flight high-resolution mass spectrometry (HPLC-QTOF). Samples were dissolved in LCMS-grade methanol (100 µg/mL) and analyzed using the SCIEX^®^ X500R QTOF LCMS system equipped with an autosampler and Infinity Lab Poroshell 120 EC-C18, 4.6 × 150 mm, 4 µm, analytical LC column. The LC was obtained under the following conditions: a gradient solvent system of acetonitrile/H_2_O with 0.05% formic acid starting at 2% acetonitrile increasing gradually to 98% over 25 min, held at 98% for 2 min, and reverted to 2% for 3 min. The injection volume and flow rate were 10 µL and 0.7 mL/minute, respectively. The MS was obtained by scanning with time-of-flight mass spectrometry (TOFMS) in a positive polarity mode at a scan range of 100 and 2000 Da. The collision energy and ion spray voltage were set at 10 V and 5500 V, respectively.

#### 4.2.5. NMR Structural Elucidation

NMR data were acquired using a 400 MHz Bruker Avance III HD Nanobay NMR spectrometer (Bruker, Billerica, MA, USA) equipped with a 5 mm BBI probe at 298 K using standard 1D and 2D NMR pulse sequences. The pulse sequences employed for the 1D spectra included the zg30 sequence using a spectral width of 8 kHz, 64 transients, a relaxation delay of 1 s, and a 90° pulse of 7. 6 µs. The 2D pulse sequences employed included the hsqcedetgpsisp2.3 and hmbcetgpl3nd pulse programs to acquire the HSQC and HMBC NMR data. The sweep widths were optimized to δH 6.00 ppm and either δC 165.00 (HSQC) or δC 220.00 (HMBC). The latter heteronuclear experiments were additionally performed with non-uniform sampling (NUS) using 25% (HSQC) or 32% (HMBC) sparse sampling. HSQC and HMBC spectra were acquired with 24 scans with 400 or 800 transients, respectively. Delay times for the 2D spectra were optimized and varied between 1.5 and 2.0 s. The HMBC spectra were collected for 1J H-C coupling constants between 120 and 170 Hz and for long-range couplings at 8 Hz. The NMR samples were prepared in deuterated solvents, and the chemical shifts referenced deuterated solvent peaks (CDCl_3_ δH 7.25, δC 77.00) and were reported in ppm.

### 4.3. TEM

#### 4.3.1. *M. tuberculosis* Growth Conditions and Treatment for Electron Microscopy

The *M*. *tuberculosis* culture was prepared as described in Section 4.2.2. The culture was grown for 10 days and adjusted to 0.5 McFarland’s standard. The cells were treated with sub-fraction 13.6 (gymnopilene) (62.5 µg/mL) or isoniazid (7.81 µg/mL), or left untreated for 3 days at 37 °C.

#### 4.3.2. TEM Sample Processing

The ultrastructural changes induced by the compound were observed using transmission electron microscopy (TEM) as previously described by Didloff et al. [21] with slight adjustments. *M*. *tuberculosis* cells were pelleted by centrifugation using Eppendorf minispin (8000 rpm/4293× *g* for 5 min) and pre-fixed with 2.5% glutaraldehyde at 4 °C overnight. After overnight fixation, the cells were centrifuged, and the fixative was removed and washed three times with 0.1 M sodium phosphate buffer (pH 7.2). Thereafter, the samples were fixed in 1% osmium tetroxide for 1 h at room temperature. The osmium tetroxide was removed, and the cells were washed three times with 0.1 M sodium phosphate buffer (pH 7.2) for 5 min.

The samples were dehydrated using a graded ethanol series (30% twice for 5 min, 50% twice for 10 min, 75% three times for 10 min each, and 100% twice for 10 min each). This was followed by acetone dehydration, starting with 1:2 acetone/ethanol for 10 min, followed by 2:1 acetone/ethanol for 15 min, and finally, 100% acetone for 10 min. After dehydration, the samples were infiltrated with Spurr’s Low-Viscosity resin as follows: 1:2 resin in acetone for 45 min, 1:1 resin in acetone for 2 h, and 3:1 resin in acetone overnight at room temperature. The 3:1 resin in acetone was removed from the samples and replaced with a 100% resin. The sample pellets and resin were transferred to labeled capsules and left to stand for 5 h at room temperature. The samples were then polymerized at 60 °C for 16–20 h.

Ultrathin sections (70 nm) were made with an ultramicrotome and placed onto 300 mesh TEM Cu grids. The sections were stained with uranyl acetate for 2 min and lead citrate (0.1 mL 10 N NaOH and 0.04 g lead citrate in 10 mL ddH_2_O) [57] for 1 min. Sections were viewed and imaged using a transmission electron microscope (JEOL JEM 2100, USA).

### 4.4. In Vitro Toxicity Screening

#### 4.4.1. Cell Culture Conditions and Cell Maintenance

Vero cells were purchased from Cellonex, RSA, and routinely maintained in 10 cm culture dishes (60.8 cm^2^) containing low-glucose DMEM supplemented with 10% FBS and 1% penicillin/streptomycin. C3A cells were purchased from the ATCC and maintained using EMEM supplemented with 10% FBS, 1% NEAA, and 1% penicillin/streptomycin. The cells were maintained in standard cell culturing conditions at 37 °C in a humidified 5% CO_2_ incubator and monitored for contamination using an Axiovert 40C inverted phase-contrast microscope (Carl Zeiss, Oberkochen, Germany). Cells were sub-cultured when they reached 70–80% confluence using the trypsinization technique described by Freshney [58].

Seeding density was determined using the trypan blue exclusion assay along with the LUNA™ Automated Cell Counter (Logos Biosystems, Republic of Korea). The trypan blue exclusion assay also allows the monitoring of cell viability. Viable cells with intact cell membranes will exclude the dye, whereas dead cells with damaged cell membranes will be unable to prevent the entry of the dye and stain blue [59].

#### 4.4.2. Image Acquisition and Data Analysis

Fluorescence micrographs were captured using an ImageXpress^®^ Micro XLS widefield High-Content Analysis System (Molecular Devices). The spatial distribution for acquisition was set at nine sites per well (3 × 3), and images were acquired using a 10× objective. With these settings, 70% of the total well area in the 96-well plate could be imaged. The filter sets used for each assay depended on the relevant dye(s) used for the assay (Table 4). The exposure time was optimized for each dye and varied for each experiment based on staining intensity.

Micrographs were analyzed using MetaXpress^®^ version 6.1 High-Content Image Acquisition and Analysis Software. The Multiwavelength Cell Scoring module was used for all experiments. Using the various analysis modules, numerous multi-parameter measurements could be obtained, including cell counts, nuclei mean area, and wavelength-specific intensities, based on segmentation parameters defined for each wavelength.

#### 4.4.3. In Vitro Cytotoxicity

Cytotoxicity screening was performed using a Hoechst 33342 and PI dual staining method [60]. Vero and C3A cells were seeded into TPP-tissue-culture-treated 96-well plates at a density of 5000 cells/100 μL/well. The cells were incubated overnight at 37 °C in a humidified 5% CO_2_ incubator to allow attachment. The cells were treated for 48 h by the addition of 100 μL aliquots of the macrofungal extract (12.5–200 μg/mL), isolated fractions/compound (3.13–50 μg/mL), isoniazid (0.63–10 mM), and the positive control melphalan (3.13–50 μM). After incubation, the treatment medium was gently aspirated and replaced with 100 μL of Dulbecco’s phosphate-buffered saline (DPBS) with Ca2^+^ and Mg2^+^ containing Hoechst 33342 (5 μg/mL) and incubated at 37 °C for 30 min. Prior to image acquisition, PI (110 μg/mL stock) was added using 10 μL aliquots to achieve a final concentration of 10 μg/mL. After image analysis, the number of live cells (nuclei stained with Hoechst only) and dead cells (nuclei stained with Hoechst and PI) was logged. Live cell numbers were used to calculate IC_50_ values from log-dose–response curves using GraphPad Prism version 5.01.

#### 4.4.4. Hepatotoxicity

Hepatotoxicity refers to liver dysfunction or damage, which is associated with an overload of drugs or xenobiotics. Hepatotoxicity was investigated using CellRox orange (oxidative stress) and TMRE/MTG (mitochondrial membrane potential and mitochondrial content) staining.

##### Production of ROS

C3A cells were seeded and treated as described for in vitro cytotoxicity (Section 4.4.3). Melphalan (50 μM) was used as a positive control. For staining, the treatment medium was aspirated and replaced with 100 μL aliquots of DPBS with Ca^2+^ and Mg^2+^ containing CellRox Orange (2.5 μM) and Hoechst 33342 (5 μg/mL). Cells were incubated at 37 °C for 30 min prior to imaging [26].

##### Mitochondrial Content and Mitochondrial Membrane Potential

C3A cells were seeded and treated as described for in vitro cytotoxicity (Section 4.4.3). Melphalan (50 μM) and carbonyl cyanide m-chlorophenylhydrazone (CCCP; 25 μM) were included as positive controls. Cells were exposed to the positive control CCCP 1 h before staining. For staining, the treatment medium was aspirated and replaced with 100 μL of DPBS with Ca^2+^ and Mg^2+^ containing TMRE (0.25 μM), MitoTracker Green (0.02 μM), and Hoechst 33,342 (5 μg/mL). The cells were incubated in the dark at 37 °C for 30 min prior to imaging [26].

#### 4.4.5. Statistical Analysis

All experiments were performed in triplicate and repeated three independent times unless otherwise stated. The standard deviation (SD) of the mean of three independent experiments was calculated and is represented by the error bars. The statistical significance of treatments compared to the untreated control was determined using the Student’s *t*-test for two samples, assuming equal variance. The data were deemed significant when * *p* ≤ 0.05, ** *p* ≤ 0.005, and *** *p* ≤ 0.001.

### 4.5. Zebrafish Embryo/Larvae Toxicity Screening

#### 4.5.1. Zebrafish Maintenance (Care), Breeding, and Embryo Collection

Methods for zebrafish maintenance, breeding, and embryo collection were adapted from Gaur et al. [60] and Murugesu et al. [61]. Adult wild-type zebrafish were purchased from an aquatic supplier, Die Kraaines Pet Warehouse, Despatch, Eastern Cape, South Africa. The zebrafish were maintained in 60 L aquarium tanks at 26.5–28.5 °C with a 14-h-light/10-h-dark cycle. Female and male zebrafish were separated to avoid spontaneous breeding. The water quality was assessed weekly to monitor the pH and levels of chlorine, nitrate, and nitrites using Tetra water quality 6-in-1 test strips. The zebrafish were fed twice daily with standard commercial dry fish feed supplemented with bloodworms during the conditioning/priming of fish for a week before breeding. For breeding, the zebrafish were placed in a breeding tank at a 1:2 male/female ratio a day before breeding to allow the zebrafish to acclimate to the environment. The specialized breeding tank (1.7 L; Techniplast, Lane Cove West, NSW, Australia) contained a separator to prevent the predation of embryos by adult zebrafish. Breeding commenced at first light, and adult zebrafish were removed and placed into the 60 L aquarium tanks. The embryos were collected from the breeding chamber by siphoning using a Pasteur pipette. The collected embryos were washed with 1× embryo medium (8.25 mL of 60X stock in 500 mL water; 60X stock: 8.7 g NaCl, 0.4 g KCl, 1.45 g CaCl_2_.2H_2_O and 2.445 g MgCl_2_.6H_2_O in 500 mL water, pH 7.2). The embryos were transferred to 10 cm culture plates containing embryo media and incubated at 28 °C for 24 h. The larvae at 24 hpf were used for subsequent experiments.

#### 4.5.2. Embryo Exposure to Extract

At 24 hpf, healthy embryos were randomly transferred to a 96-well plate (one embryo per well) in 50 µL of embryo medium using an adapted method reported by Thiagarajan et al. [62]. The embryos were exposed to various concentrations of the macrofungal extract (*G*. *junonius*: 10–80 µg/mL), which was added to the wells in 50 µL aliquots prepared in embryo medium. Five replicates were performed for both the control and treatment groups. Following treatment, the embryos were incubated at 28 °C for 72 h. The treatment of embryos was conducted using a semi-static method, with the renewal of each treatment every 24 h. This involved carefully aspirating the treatment medium to avoid damage to the embryo/larvae and replacing it with 100 µL of fresh treatment medium. Three independent experiments were conducted, and the zebrafish larvae were euthanized using tricaine (4 mg/mL) on the last day of evaluation.

#### 4.5.3. Evaluation of Morphology and Embryotoxicity

Morphology and development were evaluated every 24 h and scored based on a scoring system as described by Beekhuijzen et al. [28]. Embryos/larvae were assessed using a stereo microscope and Axiovert 40C inverted microscope (Carl Zeiss, Oberkochen, Germany). The scoring of each developmental hallmark was relative to the stage of development and compared to the control embryos. The developmental hallmarks assessed included movement, blood circulation, heartbeat, somite formation, the notochord, eye development, pigmentation of the head and body, pigmentation of the tail, the pectoral fin, and hatching. Specific scores were assigned to various developmental endpoints at different time points as described by Beekhuijzen et al. [28]. Malformation or teratogenic effects were recorded as either present or absent. Teratogenic endpoints included malformations of the head, sacculi/otolith, tail, and heart, a deformed body shape, and yolk deformation.

## Figures and Tables

**Figure 1 antibiotics-14-00179-f001:**
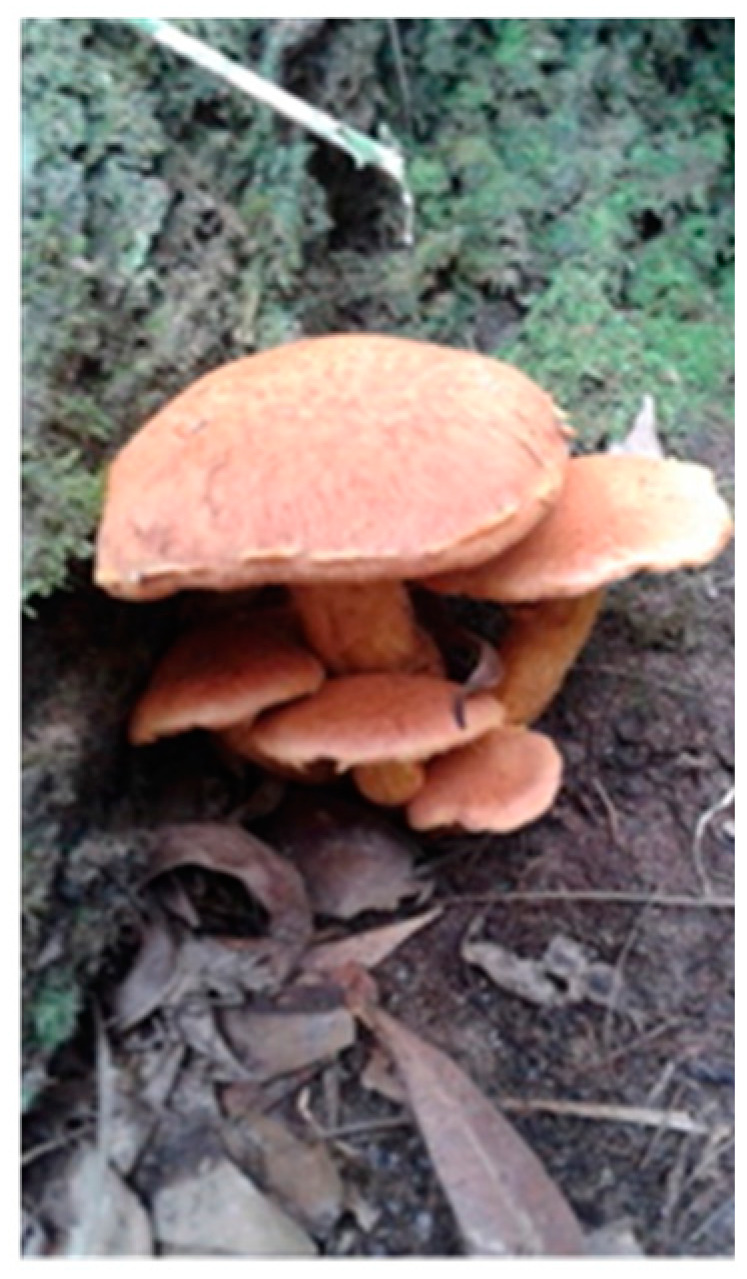
Photograph of *Gymnopilus junonius* (Fr.) P.D. Orton collected in the Western Cape province of South Africa [16].

**Figure 2 antibiotics-14-00179-f002:**
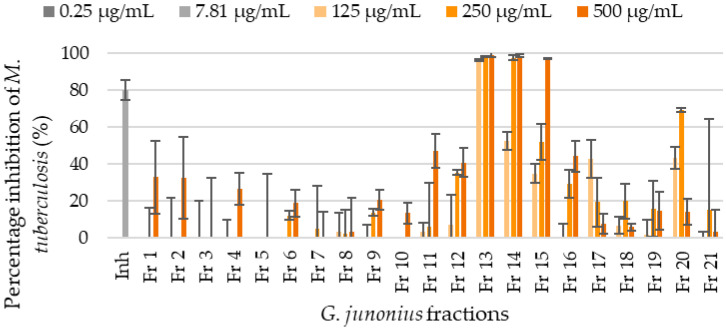
Percentage inhibition of *M*. *tuberculosis* H37 determined by the REMA for fractions isolated from an ethanol *G*. *junonius* extract using silica gel column chromatography. Data are reported as the mean ± standard deviation of triplicate readings. Inh: isoniazid.

**Figure 3 antibiotics-14-00179-f003:**
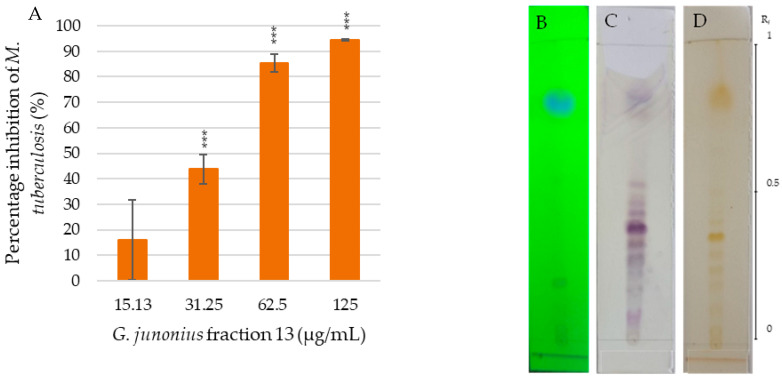
(**A**) Percentage inhibition of *M*. *tuberculosis* H37 treated with *G*. *junonius* fraction 13 determined by REMA. Data are reported as the mean ± standard deviation of three independent experiments, each performed in triplicate. Significance was determined using the two-tailed Student’s *t*-test: *** *p* ≤ 0.005 compared to the growth control; silica gel TLC profiling of fraction 13 using an ethyl acetate/methanol (9:1 *v*/*v*) solvent system with five drops of ammonium hydroxide. UV lamp wavelength (**B**) 254 nm; (**C**) staining with a vanillin–sulfuric acid solution; (**D**) iodine vapor.

**Figure 4 antibiotics-14-00179-f004:**
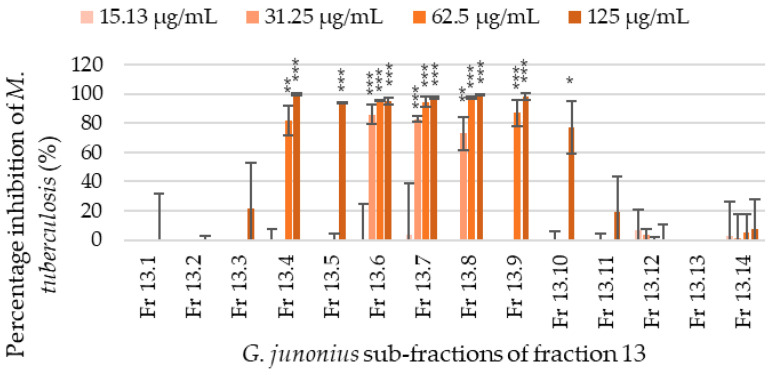
Percentage inhibition of *M*. *tuberculosis* H37 treated with *G*. *junonius* sub-fractions isolated from bioactive fraction 13 determined by REMA. Data are reported as the mean ± standard deviation of two independent experiments, each performed in triplicate. Significance was determined using the two-tailed Student’s *t*-test: * *p* ≤ 0.05; ** *p* ≤ 0.01; *** *p* ≤ 0.005 compared to the growth control.

**Figure 5 antibiotics-14-00179-f005:**
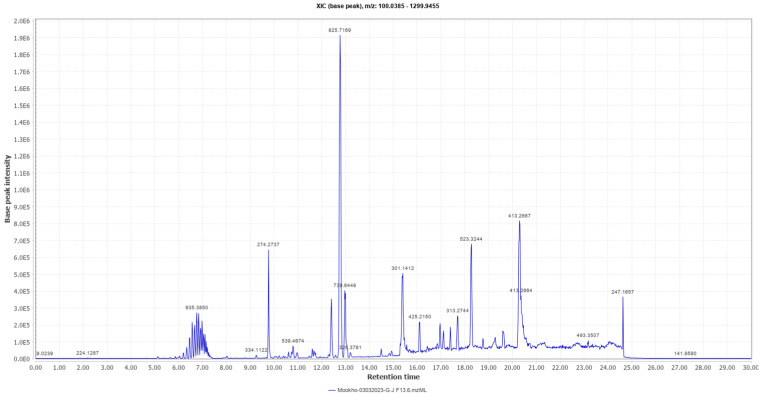
HPLC-HRESI-MS chromatogram of sub-fraction 13.6 in positive ionization mode.

**Figure 6 antibiotics-14-00179-f006:**
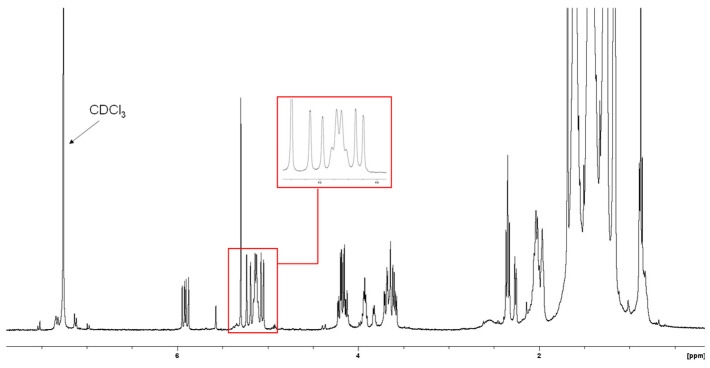
^1^H NMR spectrum (CDCl_3_, 400 MHz) of sub-fraction 13.6.

**Figure 7 antibiotics-14-00179-f007:**
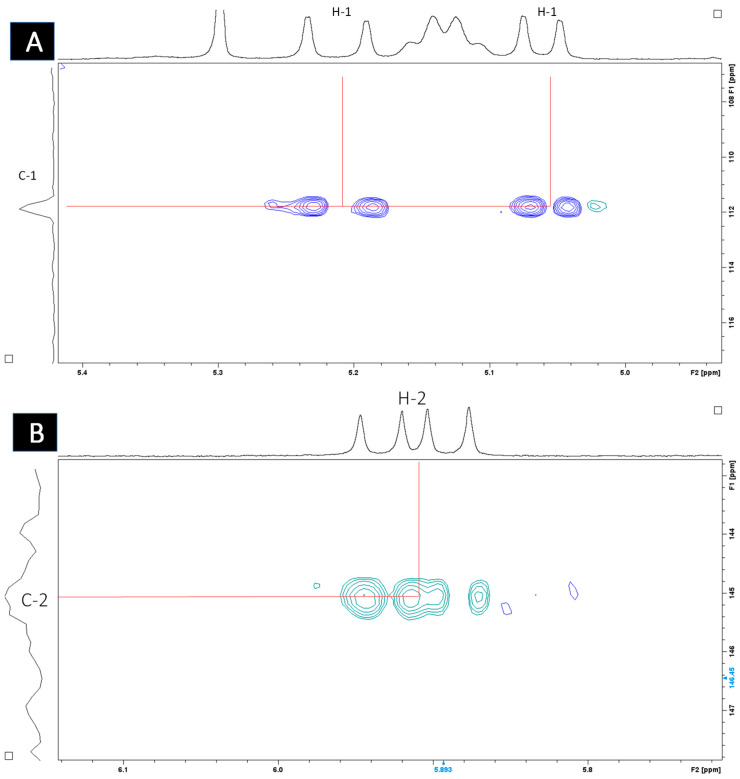
HSQC spectrum (CDCl_3_, 400 MHz) of sub-fraction 13.6 showing resonances for (**A**) H1 and (**B**) H2 vinyl protons.

**Figure 8 antibiotics-14-00179-f008:**
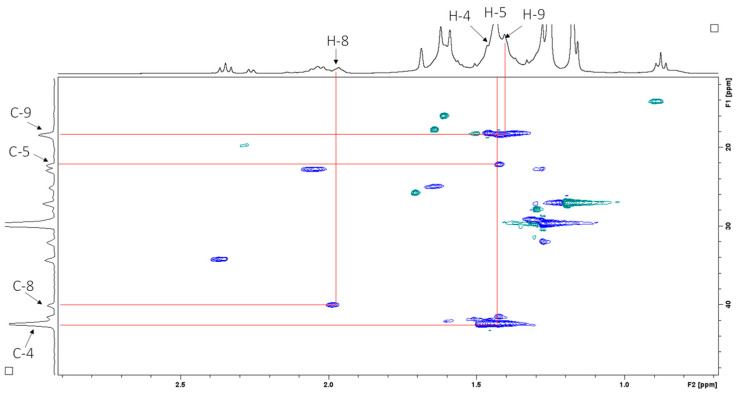
HSQC spectrum (CDCl_3_, 400 MHz) of sub-fraction 13.6 showing resonances for the aliphatic region.

**Figure 9 antibiotics-14-00179-f009:**
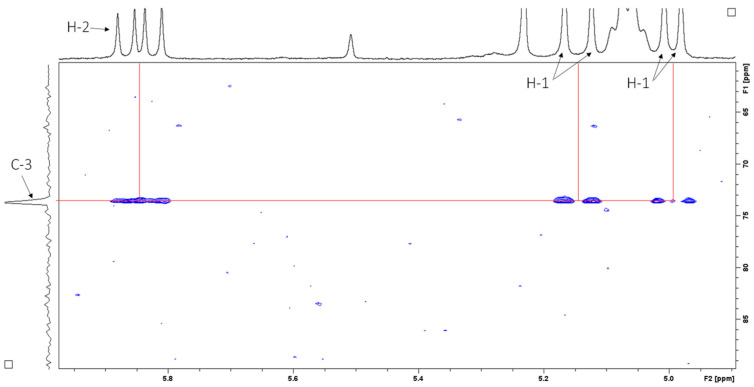
^1^H–^13^C HMBC NMR spectrum (CDCl_3_, 400 MHz) of sub-fraction 13.6 showing resonances for the vinyl protons.

**Figure 10 antibiotics-14-00179-f010:**
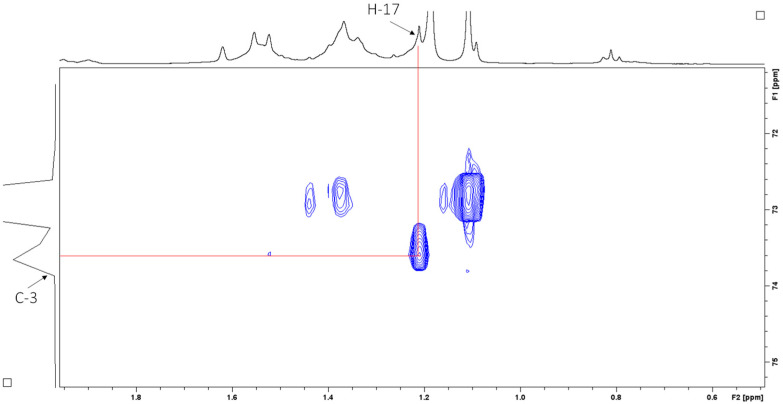
^1^H–^13^C HMBC NMR spectrum (CDCl_3_, 400 MHz) of sub-fraction 13.6 showing resonances for the aliphatic region.

**Figure 11 antibiotics-14-00179-f011:**
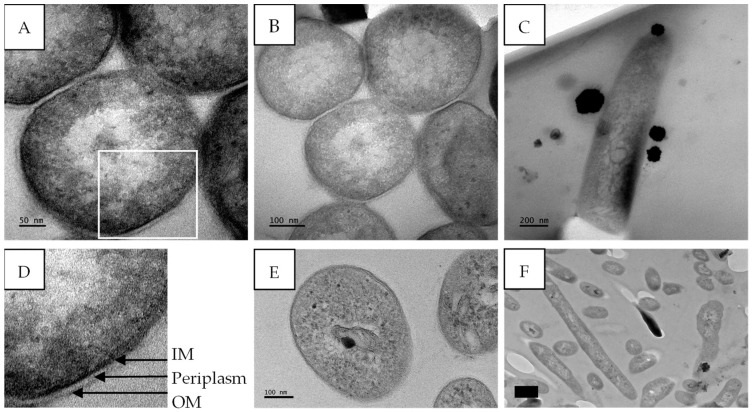
Transmission electron micrographs of untreated *M*. *tuberculosis* showing intact cell membranes. (**A**,**B**) Transverse section and (**C**) longitudinal section of *M*. *tuberculosis* cells. The outer membrane (OM), periplasm, and inner membrane/plasma membrane (IM) are visible in (**D**) (enlarged image of A). (**E**,**F**) Transverse and longitudinal sections of untreated *M*. *tuberculosis*; micrographs from [21]. Scale bar = 50–200 nm.

**Figure 12 antibiotics-14-00179-f012:**
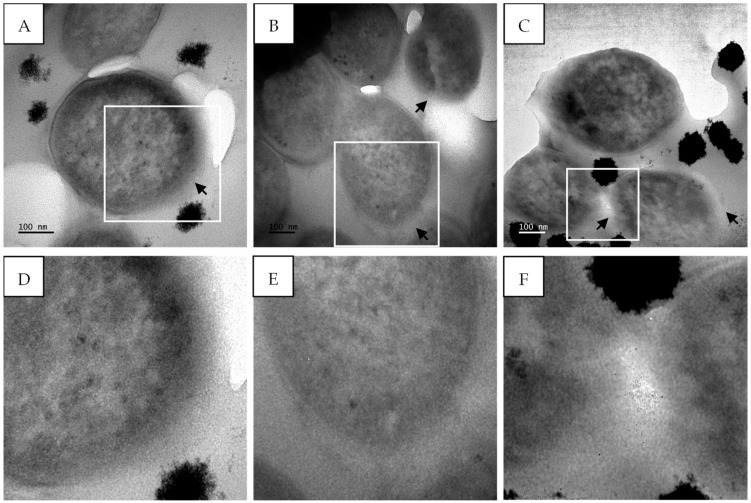
Transmission electron micrographs of *M*. *tuberculosis* showing cell membrane damage after treatment with isoniazid (7.81 µg/mL). (**A**–**C**) *M*. *tuberculosis* after treatment with isoniazid and (**D**–**F**) enlarged images of (**A**–**C**) highlighting membrane damage. Black arrows—damaged cell wall. Scale bar = 100 nm.

**Figure 13 antibiotics-14-00179-f013:**
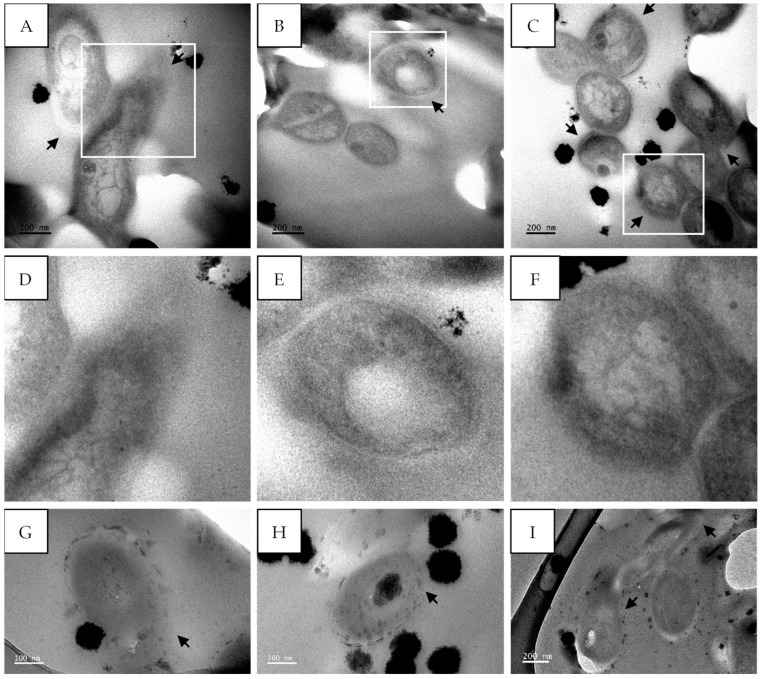
Transmission electron micrographs of *M*. *tuberculosis* showing cell membrane damage after treatment with sub-fraction 13.6 containing the predominant compound, gymnopilene (62.5 µg/mL), isolated from an ethanol extract of *G*. *junonius*. (**A**–**C**,**G**–**I**). *M*. *tuberculosis* after treatment with *G*. *junonius* sub-fractions 13.6–8 and (**D**–**F**) enlarged images of (**A**–**C**) highlighting membrane damage. Black arrows—damaged cell wall. Scale bar = 100–200 nm.

**Figure 14 antibiotics-14-00179-f014:**
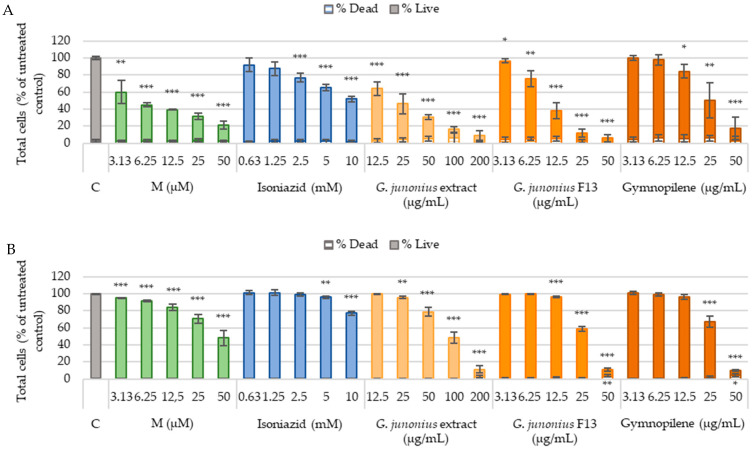
Cytotoxicity screening of crude macrofungal extract, isolated fraction, and predominant compound gymnopilene in sub-fraction 13.6 after 48-h exposure to C3A (**A**) and Vero (**B**) cells. The percentages of live and dead cells were determined using the Hoechst 33242 and PI dual staining method. C: untreated control; M: melphalan. Data are reported as the mean ± standard deviation of three independent experiments, each performed in triplicate. Significance was determined using the two-tailed Student’s *t*-test: * *p* ≤ 0.05, ** *p* ≤ 0.01, and *** *p* ≤ 0.005 compared to the untreated control.

**Figure 15 antibiotics-14-00179-f015:**
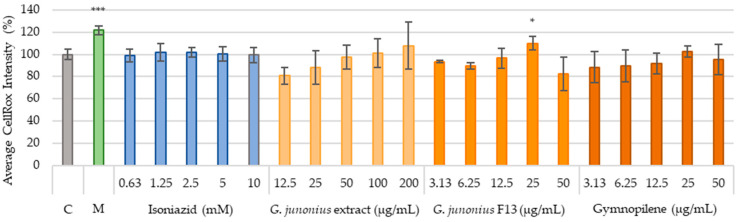
CellRox intensity observed after 48-h treatment of C3A cells with *G*. *junonius* extract, isolated fractions, and predominant compound gymnopilene in sub-fraction 13.6. Intracellular ROS accumulation was determined using the CellRox orange staining method. C: untreated control; M: melphalan (50 µM). Data are reported as the mean ± standard deviation of three independent experiments, each performed in triplicate. Significance was determined using the two-tailed Student’s *t*-test: * *p* ≤ 0.05, and *** *p* ≤ 0.005 compared to the untreated control.

**Figure 16 antibiotics-14-00179-f016:**
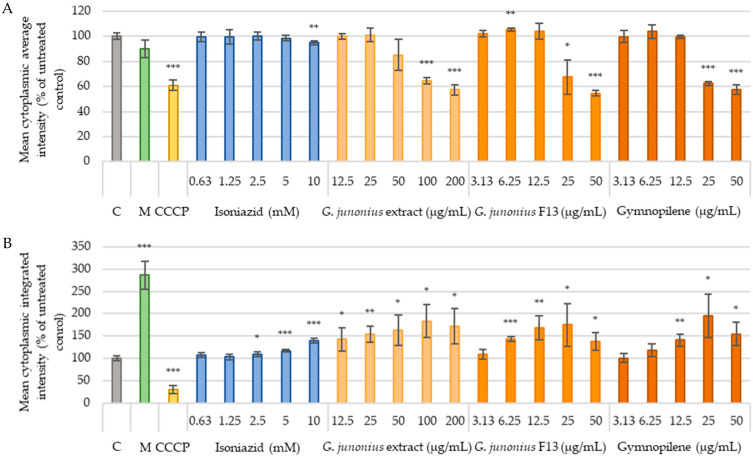
TMRE (**A**) and MitoTracker Green (**B**) intensity observed after 48-h treatment of C3A cells with *G*. *junonius* extract, isolated fraction, and predominant compound gymnopilene in sub-fraction 13.6. Mitochondrial function was determined using the TMRE/MTG staining method. C: untreated control; M: melphalan (50 µM); CCCP: carbonyl cyanide m-chlorophenylhydrazone (25 µM). Data are reported as the mean ± standard deviation of three independent experiments, each performed in triplicate. Significance was determined using the two-tailed Student’s *t*-test: * *p* ≤ 0.05, ** *p* ≤ 0.01, and *** *p* ≤ 0.005 compared to the untreated control.

**Figure 17 antibiotics-14-00179-f017:**
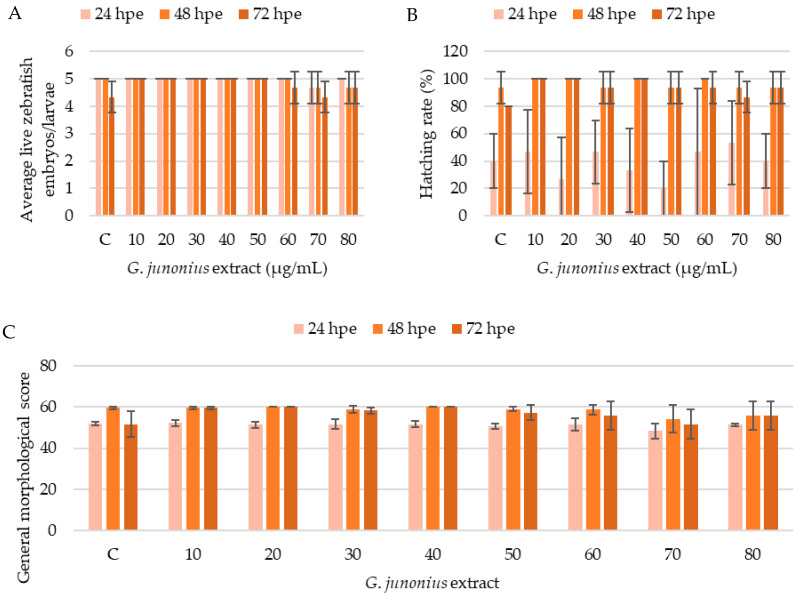
Assessment of the effects of *G*. *junonius* extract on zebrafish embryos/larvae using a modified zebrafish embryotoxicity test. (**A**) Average number of live zebrafish embryos/larvae after treatment with *G*. *junonius* extract for 24, 48, and 72 hpe; (**B**) effect of *G*. *junonius* extract on zebrafish embryo hatching; (**C**) general morphological score of zebrafish embryos/larvae. C: control. Data are reported as the mean ± standard deviation of three independent experiments.

**Figure 18 antibiotics-14-00179-f018:**
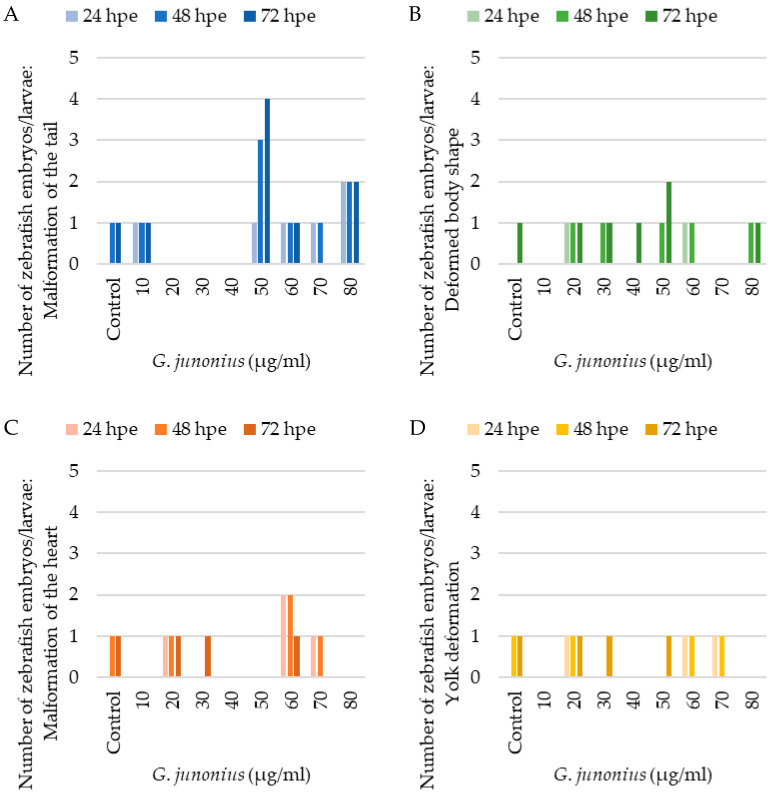
Assessment of teratogenic effects of *G*. *junonius* extract on zebrafish embryos/larvae development. (**A**) Number of zebrafish embryos/larvae displaying malformations of the tail; (**B**) deformed body shape; (**C**) malformation of the heart; and (**D**) yolk deformation after 24, 48, and 72 hpe.

**Figure 19 antibiotics-14-00179-f019:**
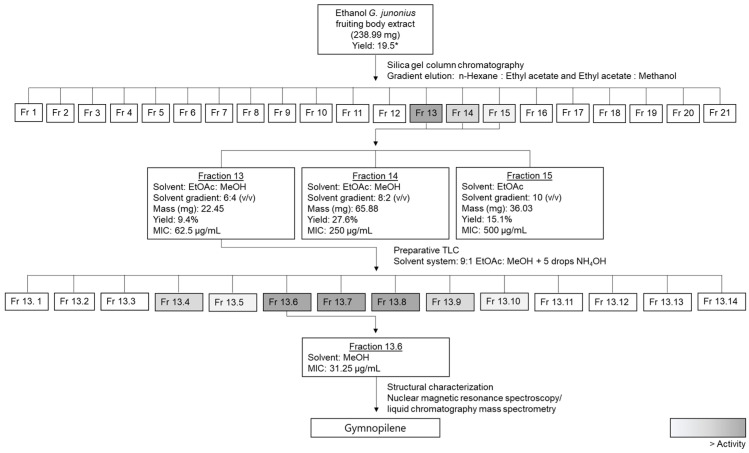
Biological-activity-guided fractionation scheme to isolate the antimycobacterial compound(s) from a *G*. *junonius* ethanol extract, highlighting the fractions with activity. Masses are representative of an individual isolation procedure. * Percentage yield reported by Boukes et al. [16].

**Table 1 antibiotics-14-00179-t001:** ^1^H (CDCl_3_, 400 MHz) and 2D NMR (HSQC and HMBC) spectroscopic data for sub-fraction 13.6 and gymnopilene reported in the literature (lit) [25].

Carbon No.	^13^C (lit), CD_3_OD@	^1^H (lit), CD_3_OD	^1^H (CDCl_3_)	^13^C (CDCl_3_) ^#^
1	112.0 (CH2)	5.20, dd, J = 17.4, 1.7 Hz 5.02, dd, 10.7, 1.7	5.21 5.06	111.6
2	146.3 (CH)	5.91, dd, J = 17.4, 10.7 Hz,	5.91	145.0
3	73.8 (C)	-	-	-
4	42.7 (CH2)	1.55–1.27, m (44H)	1.42, br m	42.2
5	23.2 (CH2)	2.00, m	2.03, br m	22.6
6	125.8 (CH)	5.10, m	5.14	124.5
7	136.0 (C)	-		
8	41.3 (CH2)	2.00, m	1.97	39.8
9	19.4 (CH2 × 7)	1.55–1.27, m (×7)	1.42, br m	18.1
10	43.4 (CH2 × 7)	1.55–1.27, m (×7)	1.42, br m	42.2
11	73.4 (C × 7)	-		
12	43.4 (CH2 × 7)	1.55–1.27, m (×7)	1.42, br m	42.2
13	23.7 (CH2)	2.00, m	2.03, br m	22.6
14	125.8 (CH)	5.10, m	5.14	124.5
15	132.0	-		
16	25.9 (CH3)	1.60–1.67, s	1.68	25.7 *
17	27.6 (CH3)	1.24, 3H, s	1.27	27.2
18	15.9 (CH3)	1.60–1.67, s	1.61	15.8 *
19	27.0 (CH3 × 7)	1.15, br s (×7)	1.17	26.9
20	17.8 (CH3)	1.60–1.67, s	1.58	17.6 *

(*) Chemical shifts may be interchangeable. (#) ^13^C NMR chemical shifts were obtained from HSQC and HMBC spectra.

**Table 2 antibiotics-14-00179-t002:** IC_50_ values (µg/mL) and selectivity indices (SI) of the crude macrofungal extract, isolated fraction, and sub-fraction 13.6 containing gymnopilene against Vero and C3A cells. IC_50_ data are reported as the mean ± standard deviation of three independent experiments, each performed in triplicate.

	*M*. *tuberculosis* MIC	Vero IC_50_	Vero SI (IC_50_/MIC)	C3A IC_50_	C3A SI (IC_50_/MIC)
Crude Extract	250 µg/mL	90.86 ± 1.04	0.36	18.93 ± 1.09	0.076
Fraction 13	62.5 µg/mL	27.02 ± 1.01	0.43	9.16 ± 1.05	0.15
Sub-Fraction 13.6 (Gymnopilene)	31.25 µg/mL	28.53 ± 1.02 (34.59 µM)	0.91	22.38 ± 1.09 (27.14 µM)	0.72

**Table 3 antibiotics-14-00179-t003:** Solvent system used in silica gel column chromatography for fractionation of ethanol macrofungal extracts.

Fraction	Hexane (mL)	Ethyl Acetate (mL)	Methanol (mL)
1	10	0	-
2	9	1	-
3	8	2	-
4	7	3	-
5	6	4	-
6	5	5	-
7	4	6	-
8	3	7	-
9	2	8	-
10	1	9	-
11	0	10	-
12	-	8	2
13	-	6	4
14	-	4	6
15	-	2	8
16–21	-	0	10

**Table 4 antibiotics-14-00179-t004:** Summary of the fluorescent dyes and filter sets used for each toxicity screening assay.

Assay	Fluorescent Dye	Filter Set	Excitation/Emission Wavelength (nm)
Cytotoxicity	Hoechst 33342 PI	DAPI Texas Red	377/477 562/624
Reactive oxygen species (ROS)	CellRox Orange	TRITC	543/593
Mitochondrial content and membrane potential	TMRE MTG	TRITC FITC	543/593 482/536

## Data Availability

The raw data supporting the conclusions of this article will be made available by the authors on request.

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
