# Peer review of "Exploring Antimycobacterial Potential: Safety Evaluation and Active Compound Isolation from Gymnopilus junonius"

_antibiotics, 2025, doi:10.3390/antibiotics14020179_

Round 1

Reviewer 1 Report

Comments and Suggestions for Authors

1. I strongly suggest removing the picture frames to make the layout of the manuscript more presentable.

2. I strongly suggest removing the grid lines at the bottom of the graphics to make the layout of the manuscript more presentable.

3. I strongly suggest standardizing the alignment of the Y-axis titles of all the graphs, to make the layout of the manuscript more presentable.

4. lines 36-38: The sentence does not add pertinent information, I suggest removing it.

5. line 88: please include a reference that supports the information presented.

6. line 98: reference format seems inconsistent with the journal.

7. lines 104-106 seems to be a more appropriate objective than the one presented in line 21-22 of the abstract, I suggest modifying the abstract.

8. Lines 101-102: Since the study, as mentioned earlier also investigated the antimicrobial activity of this same genus of macrofungus, what is the main difference between that study and this manuscript?

9. Line 673: reference format seems inconsistent with the journal.

10. line 678: at what temperature?

11. Line 693: please include a reference that supports the information presented.

12. Line 745: reference format seems inconsistent with the journal.

13. line 754: format error.

14. Line 758: format error.

15. Lines 747-745:

a. how was the mycobacterium suspension prepared?

b. Were the cells washed and collected or used directly from the liquid culture medium?

16. line 782; 915 and throughout the manuscript were is necessary: write down the meaning of this and other acronyms at least the first time they appear.

17. Lines 693;  734 (equation); 781; 801; 816; and so on:  I felt there was a lack of references to support the methodology presented. Or were the methods and procedures created and standardized by this study?

18. line 867: area?

19. line 929 and wherever else necessary: please use italics.

20. Line 947: no multiple comparisons were made? were all the comparisons in the pictures 2 by 2? If so, make this clear in the caption of all the figures. If another type of comparison was made, include the name of the statistical test here and in the figure legend.

21. line 978: how many embryos per well were used?

22. line 977 and wherever else necessary in the manuscript: replace the term embryo with lava. The term embryo is intended for the phase inside the “egg”, once hatched this is the wrong nomenclature for the manuscript.

23. lines 111-121

a. this period seems confusing. Just for clarification, the extracts used in this manuscript originate from references 20-21?!

b. the antimicrobial activity of this manuscript was done against the same strain as the original study, is that it?!

24. Figure 3B: does not add information to the manuscript, I suggest removing it.

25. What is the difference between Figure 3A and Figure 2? Are the data repeated?

26. Figure 2, 3 and 4: it's not clear how this percentage was obtained? Why not present it as “Total activity” as described in the methods?

27. Figure 14, 15: please adjust the size of the writing inside the graphics to avoid overlapping.

28. Please check that the number of repetitions and replicates for each experiment is the same in the figures and tables against the figure and table legends.

29. Lines 408-409: reference format seems inconsistent with the journal.

30. lines 408-410: please describe this in detail in the methods section.

31. Figure 17: Why aren't the deviations shown? And no statistical analysis?

Comments on the Quality of English Language

The text needs to be revised and the English improved. There are also formatting errors throughout the text.

Reviewer 2 Report

Comments and Suggestions for Authors

The manuscript explores the antimycobacterial activity of gymnopilene, isolated from Gymnopilus junonius, against Mycobacterium tuberculosis. While the study is intriguing and holds promise, several enhancements are necessary to improve the manuscript for publication.

Firstly, the chemical formula C50H96O8 should be revised to adhere to scientific norms, ensuring that the numbers are correctly written in subscript, as C₅₀H₉₆O₈. This adjustment will ensure clarity and accuracy in the chemical representation.

Additionally, the Materials and Methods section, particularly paragraphs 4.4 and 4.5, requires the inclusion of appropriate references to support the methodologies used. Providing these references will enhance the credibility of the experimental procedures and align the manuscript with standard scientific practices.

A dedicated Conclusions section is recommended. This section should succinctly summarize the key findings, discuss the study's limitations, and outline potential directions for future research. Including these elements will provide a comprehensive closure to the study and guide readers on the implications and next steps.

Lastly, the citation style within the manuscript needs to be consistent. Currently, citations appear in multiple formats, such as superscript, (author name, year), or in square brackets. The manuscript should be revised to follow a uniform citation style that aligns with the journal's guidelines, ensuring a professional and cohesive presentation.

Reviewer 3 Report

Comments and Suggestions for Authors

This study isolated gymnopilene from *Gymnopilus junonius* ethanol extract, showing antimycobacterial activity (MIC: 31.25 µg/mL) against *Mycobacterium tuberculosis*. Gymnopilene caused cell wall disruption, demonstrated cytotoxicity in mammalian cells, but was safe in zebrafish larvae, highlighting macrofungi's pharmaceutical potential. 

The authors conducted a thorough and in-depth investigation. However, to further enhance the practical applicability and impact of this study, it would be highly beneficial to include detailed information on the yield of each critical step in the extraction and fractionation process. Specifically, providing the yields of the crude extract, Fraction 13, and Sub-Fraction 13.6 would be valuable. This data would serve as a crucial reference for industries aiming to explore the potential of these bioactive compounds for commercial development. By understanding the efficiency and scalability of the process, industries can better assess the feasibility of advancing towards large-scale production and eventual commercialization.

Round 2

Reviewer 1 Report

Comments and Suggestions for Authors

The authors responded satisfactorily to all my comments. Thank you.

Reviewer 2 Report

Comments and Suggestions for Authors

Agree with revised manuscript